# Self-Influence Guided Data Reweighting for Language Model Pre-training

**Megh Thakkar**[1*]    **Tolga Bolukbasi**[2]    **Sriram Ganapathy**[3,4]
**Shikhar Vashishth**[3]    **Sarath Chandar**[1,5,6]    **Partha Talukdar**[3]

[1]Mila – Quebec AI Institute    [2]Google Deepmind    [3]Google Research India
[4]Indian Institute of Science    [5]Polytechnique Montréal    [6]Canada CIFAR AI Chair
{megh.thakkar,sarath.chandar}@mila.quebec
{srigana,shikharv,tolgab,partha}@google.com

## Abstract

Language Models (LMs) pre-trained with self-supervision on large text corpora have become the default starting point for developing models for various NLP tasks. Once the pre-training corpus has been assembled, all data samples in the corpus are treated with equal importance during LM pre-training. However, due to varying levels of relevance and quality of data, equal importance to all the data samples may not be the optimal choice. While data reweighting has been explored in the context of task-specific supervised learning and LM fine-tuning, *model-driven* reweighting for pre-training data has not been explored. We fill this important gap and propose PRESENCE, a method for jointly reweighting samples by leveraging self-influence (SI) scores as an indicator of sample importance and pre-training. PRESENCE promotes novelty and stability for model pre-training. Through extensive analysis spanning multiple model sizes, datasets, and tasks, we present PRESENCE as an important first step in the research direction of sample reweighting for pre-training language models.

## 1  Introduction

Language models (LM), typically pre-trained on large volumes of unlabeled text data, have become ubiquitous model choices for various challenging downstream tasks  (Lewkowycz et al., 2022; Driess et al., 2023). The fundamental direction pursued for improving language model pre-training involves increasing the amount of training data or scaling model size (Scao et al., 2022). The training data is generally assembled from scraping the web and filtered using manually crafted heuristics that often require domain expertise (Xue et al., 2021). A key similarity in these prior works is the uniform treatment of the data samples available in the assembled corpora, without any regard for the data quality.

Prior works for both model-based sample selection (Swayamdipta et al., 2020) and reweighting (Mindermann et al., 2022) use a supervised learning setup. They often rely on curating special validation sets (Jain and Shenoy, 2022), proxy models (Pleiss et al., 2020; Mindermann et al., 2022), or utilizing loss and prediction uncertainty signals based on ground-truth labels (Kawaguchi and Lu, 2020; Coleman et al., 2020). Adaptation of these methods to pre-training is often non-trivial. Performance of a pre-trained model on downstream tasks cannot be predicted by its pre-training validation performance. Moreover, offline filtering using proxy models is quite expensive for the massive scale of pre-training data (Liu et al., 2022).

In this paper, we attempt to develop an effective data reweighting framework for language model pre-training. We use *self-influence* (SI), the degree to which a given training sample affects model training and its own prediction, as an indicator of sample importance for pre-training. SI scores have been previously shown to be effective in identifying noisy and outlier samples (Yeh et al., 2018), but these evaluations have been limited to supervised settings. We first verify the ability of SI scores to predict sample quality of pre-training data, such as noisy text and domain mismatched samples. We then probe their effectiveness for pre-training dataset selection by using them to filter out noisy samples in the pre-training data derived from the web.

Based on our analysis which shows that self-influence scores can be used as an indicator of sample importance, we propose PRESENCE: **P**re-training data **Re**-weighting with **S**elf-influ**ence**. PRESENCE is an online and adaptive data reweighting method that uses self-influence scores to weigh samples in a training batch. We note that during pre-training, the training loss decreases exponentially in the initial steps, with a minimal decrease in loss values in the subsequent stages (Yang

---

*Work done while at Google Research India

et al., 2021). Furthermore, well-trained models can identify noisy samples better when used to calculate SI scores, as compared to models in very early stages of training (Pruthi et al., 2020). Based on these observations, we formulate a two stage reweighting strategy: (i) in the first stage of learning, data samples with higher SI scores are emphasized more to drive learning using influential samples, while (ii) in the second stage, data samples with higher SI scores are de-emphasized. This limits the impact of noisy and unreliable samples while giving more weight to the higher quality samples. To the best of our knowledge, this is the first work that evaluates the use of influence functions for sample selection and reweighting at the scale of pre-training. Our contributions are as follows:

- We initiate a study into data reweighting for pre-training and establish the relationship of self-influence (SI) scores with sample characteristics such as noise and domain mismatched information in the training data.

- We present sequential data filtering using SI scores as an effective data selection strategy for pre-training, and evaluate the performance of models pre-trained on large-scale filtered datasets. We call this method PRESENCE-Sequential.

- Building on our findings, we propose PRESENCE, a model-driven sample reweighting method using self-influence scores that jointly weighs samples and enables learning. PRESENCE promotes novelty and stability for model pre-training.

- Through extensive experiments and analyses spanning multiple model sizes, datasets, and tasks, we demonstrate that PRESENCE provides consistent gains over pre-training using randomly sampled pre-training corpora or SI score based filtered data. We believe PRESENCE is an important step in the research direction of data sample weighting for pretraining.

## 2 Background: TracIn and Self-influence

Though PRESENCE can be used with any influence function, we use TracIn (Pruthi et al., 2020) based self-influence score due to its scalability, generalizability, and effectiveness in identifying outliers.

**Self-influence using TracIn:** TracIn computes *influence*, i.e., how the loss on the test point changes during the training process whenever the training sample of interest was utilized by a first-

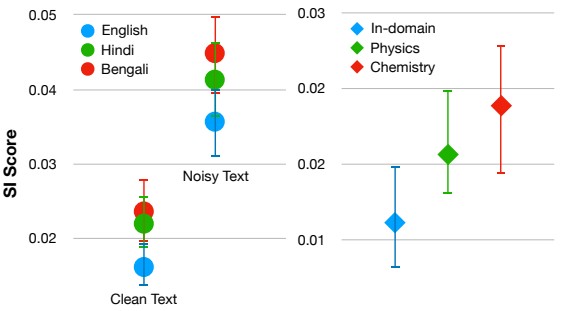

Figure 1: **Left** - Average SI scores across English, Hindi, and Bengali for clean and noisy (jumbled) text; **Right** - Average SI scores for randomly sampled and domain mismatched information in English. We observe high SI scores on average for noisy and domain mismatched text. More discussions in Section 2.1.

order gradient approximation. For a model $f$ with parameters $\theta$ and loss function $l(f_\theta, \cdot)$, the gradient $g(\theta, \cdot)$ for a sample $z$ is $g(f_\theta, z) = \nabla l(f_\theta, z)$. The TracIn$(f_\theta, \cdot, \cdot)$ *influence* of training sample $z$ on test sample $z'$ is given by,

$$\text{TracIn}(f_\theta, z, z') = g(f_\theta, z) \cdot g(f_\theta, z') \tag{1}$$

*Self-influence* score measures the *influence* a sample has on itself. This is identical to replacing $z'$ with $z$ in Equation 1, giving TracInSI$(f_\theta, \cdot)$ as,

$$\text{TracInSI}(f_\theta, z) = g(f_\theta, z) \cdot g(f_\theta, z) \tag{2}$$

### 2.1 Relationship between Self-Influence and Sample Quality

We investigate the relationship between self-influence (SI) scores and sample quality by probing noisy and domain mismatched samples. We expect these samples to have high self-influence scores as they tend to reduce the loss w.r.t. a well-trained model (Yeh et al., 2018). We use a pre-trained mT5-base (Xue et al., 2021) model and calculate self-influence with a span-corruption loss. We randomly sample 10,000 samples for three languages in mC4 (Raffel et al., 2020) and calculate the self-influence scores of *clean* or original samples and their corresponding *noisy* samples, i.e., samples with a permuted word order. Similarly, we calculate average self-influence scores over domain mismatched samples and compare them with average scores over randomly sampled English corpus.

As shown in Figure 1, we observe substantially high average self-influence scores for noisy samples across all languages as well as for domain mismatched text in English. The results indicate

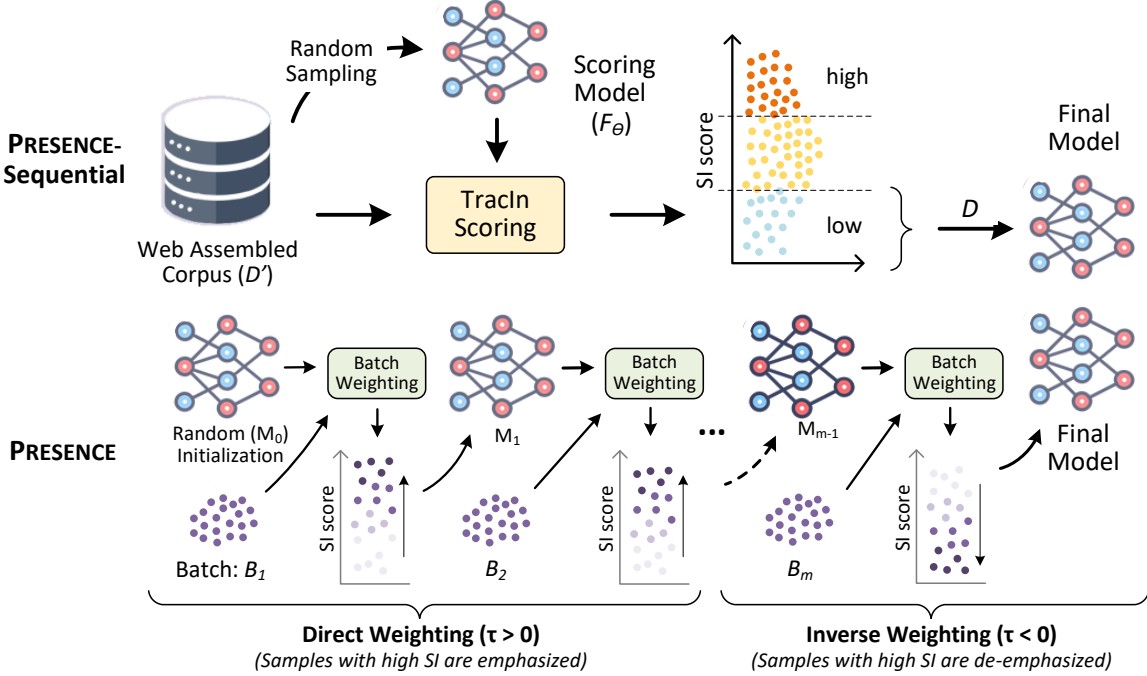

Figure 2: Overview of PRESENCE-Sequential (top, Section 3) and PRESENCE (bottom, Section 4.1). PRESENCE-Sequential filters out data in a sequential manner, first training a scoring model and then using it to filter data. PRESENCE is a joint sample reweighting strategy that leverages SI scores for sample weighting within a minibatch.

that SI scores can be used to distinguish between correct and noisy text and they can also be used to detect data from a novel domain.

## 3 PRESENCE-Sequential: Filtering Pre-training Data using Self-Influence

**Extending TracIn based Self-influence for Pre-training:** As pre-training is computationally expensive, we leverage the layer agnostic nature of TracIn and introduce an optimized layer-wise self-influence calculation. For layers $K = \{k_1, k_2, \ldots, k_K\}$ of model $f_\theta$, let $f_{\theta,k}$ denote the parameters of layer $k$. Self-influence for any layer set $\mathcal{K} \subset K$ is,

$$\text{TracInSI}_{\mathcal{K}}(f_\theta, z) = \sum_{k \in \mathcal{K}} \text{TracInSI}(f_{\theta,k}, z) \quad (3)$$

As shown in Section 2.1, there is a relation between SI scores and the sample quality. We leverage this property to filter large-scale web corpora in an offline manner to create more suitable pre-training data. We present an overview of our offline sequential filtering strategy using self-influence scores, called PRESENCE-Sequential, in Figure 2. Assuming that a model requires $N$ training samples for pre-training, we choose $N$ samples from a set of $N' > N$ samples by filtering out samples with the highest SI scores using a proxy model trained

| Dataset | Task | # Languages | Metric |
|---------|------|-------------|--------|
| XQuAD | Question Answering | 10 | F1 |
| MLQA | Question Answering | 7 | F1 |
| TyDi QA | Question Answering | 11 | F1 |
| XNLI | Sentence Pair | 14 | Accuracy |
| WikiAnn NER | Structured Prediction | 40 | Span-F1 |

Table 1: Datasets, tasks, # languages and metrics.

on randomly sampled data (SI Scoring Model). To obtain a relevant pre-training set $D$ ($|D| = N$), from the larger corpora $D'$ ($|D'| = N'$), we use the scoring model $F_\theta(\cdot)$ to calculate the SI scores using Equation 3 for all samples $d_i \in D'$,

$$\text{TracInSI}_{\mathcal{K}}(F_\theta, D') = \{\text{TracInSI}_{\mathcal{K}}(F_\theta, d_i | d_i \in D')\} \quad (4)$$

We sort $\text{TracInSI}_{\mathcal{K}}(F_\theta, D')$ in increasing order of SI scores and filter out $N' - N$ samples with the highest score. The remaining $N$ samples comprise the filtered set $D$ used for pre-training,

$$\begin{aligned} D = \{d_i | d_i \in D'\} \\ \forall i : i \in \text{sorted}(\text{TracInSI}_{\mathcal{K}}(F_\theta, D'))[1:N] \end{aligned} \quad (5)$$

**Pre-training Setup:** We use the mC4 dataset (Xue et al., 2021) and pre-train an mT5-base model for $200,000$ steps on randomly shuffled data, and use this as the 'Scoring Model ($F_\theta$)' to create the

| Model | Question Answering | | | Sentence Pair | Structured |
|---|---|---|---|---|---|
| | XQuAD | MLQA | TyDi QA-GoldP | XNLI | WikiAnn NER |
| Metrics | F1 | F1 | F1 | Accuracy | Span-F1 |
| *Cross-lingual zero-shot transfer (models fine-tuned on English data only)* | | | | | |
| mt5-base* | 72.33 | 61.60 | 49.09 | 69.98 | 41.42 |
| mT5-base+PRESENCE-Sequential-reverse | 70.17 | 59.72 | 47.92 | 68.64 | 37.39 |
| mT5-base+PRESENCE-Sequential | **73.40** | **61.95** | **51.64** | **71.22** | **44.63** |
| *Translate-train (models fine-tuned on English data plus translations in target languages)* | | | | | |
| mt5-base* | 78.26 | 65.45 | 52.75 | 76.76 | 80.86 |
| mT5-base+PRESENCE-Sequential-reverse | 76.83 | 64.76 | 46.01 | 75.36 | 79.14 |
| mT5-base+PRESENCE-Sequential | **78.96** | **66.04** | **57.65** | **77.74** | **81.45** |

Table 2: Performance comparison of using PRESENCE-Sequential to filter out pre-training data. PRESENCE-Sequential filters out noisy pre-training samples using SI scores and achieves better results than the baselines. **Bold** shows the best result (discussions in Section 3). * denotes our reproductions. Discussion in Section 3.1.

filtered dataset. We pre-train an mT5-base model from scratch on the filtered mC4 set for $200,000$ steps by choosing samples with the least SI scores that are theoretically more suitable for model learning. The models are trained with a batch size of $1024$, with an input token length of $1024$ and output token length of $229$. Following Raffel et al. (2020), we use a base learning rate of $1.0$ with $10000$ warm-up steps, an inverse square root learning rate decay schedule, and a loss-normalizing factor of $234496$. We use the first layer of the encoder and the first layer of the decoder in the set $\mathcal{K}$ for TracInSI$_{\mathcal{K}}$.

**Downstream Tasks and Fine-tuning:** Following Xue et al. (2021), we utilize datasets across 5 tasks from the XTREME multilingual benchmark (Hu et al., 2020), including Question Answering, Sentence-Pair, and Structured Prediction. We evaluate on (i) zero-shot cross-lingual transfer: where the fine-tuning data is only in English, and (ii) translate-train: where the fine-tuning data is in English and translated into the target languages for all the downstream datasets. We summarize the datasets used for evaluation in Table 1. We fine-tune all the models on the downstream tasks using a batch size of $128$, with a learning rate of $0.001$, and a dropout rate of $0.1$ for $20,000$ steps.

### 3.1 Results and Analysis

We compare the performance of the model pre-trained on filtered web corpora (mT5-base+PRESENCE-Sequential) with the baseline model trained on randomly sampled data in Table 2. We observe that when we filter out samples with high SI scores, we obtain consistent gains over the baseline models. This indicates that SI

scores can be used as an indicator of sample quality and can be used for pre-training dataset filtering. To further test our hypotheses, we pre-train a model on data created by removing low SI samples (reverse ranking). We label this model mT5-base+PRESENCE-Sequential-reverse. This model performs significantly worse compared to the baseline, further validating that SI scores are indeed an indicator of the sample quality, and are effective in identifying noisy samples in the large-scale pre-training corpora.

However, as mentioned, PRESENCE-Sequential requires different expensive sequential processes: (i) Pre-train a SI scoring model, and (ii) pre-train a second model on the filtered dataset. Since pre-training is computationally expensive, we explore a joint sample reweighting adaptation next.

## 4 PRESENCE: Sample Reweighting using Self-influence

In this approach, we use the SI scores in an online joint setting by reweighting samples at the minibatch level. We calculate sample SI scores at each training step and use them to weigh sample gradients before aggregating them for the gradient update. To formulate sample reweighting using SI scores, we consider batch $B = \{z_i | i \in [1, n]\}$, where $z_i$ denotes a sample. We calculate SI scores using Equation 3 for each sample to get array $\mathcal{S}$, where $|\mathcal{S}| = n$,

$$\mathcal{S} = \{s_i | s_i = \text{TracInSI}_{\mathcal{K}}(f_\theta, z_i); i \in [1, n]\} \quad , \quad (6)$$

where $s_i$ denotes the SI score of $z_i$. We normalize $\mathcal{S}$ for numerical stability and uniformity,

$$\mathcal{S} \leftarrow \text{normalize}(\mathcal{S}) = \frac{\mathcal{S} - \mu(\mathcal{S})}{\sqrt{\sigma^2(\mathcal{S}) + \epsilon}} \quad , \quad (7)$$

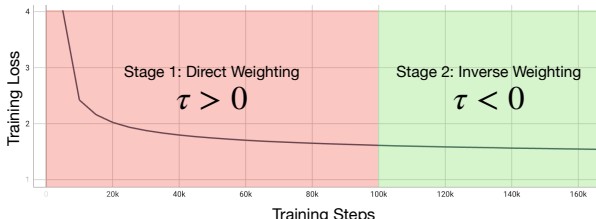

Figure 3: PRESENCE's two-stage reweighting based on the training loss. We perform *direct* weighting in the initial stage and *inverse* weighting next (Section 4.1).

---

where $\mu(\cdot)$ and $\sigma^2(\cdot)$ denote the mean and variance, respectively, and $\epsilon$ is a small number. To calculate relative weights for the samples' gradients, we use a softmax function, softmax$(\cdot)$ over each $s_i$ with temperature $\tau$ to get weights $w_i$,

$$w_i = \text{softmax}(s_i, \tau) = \frac{e^{\tau \cdot s_i}}{\sum_{s_i \in \mathcal{S}} e^{\tau \cdot s_i}} \quad (8)$$

Here, $w_i$ gives the weight for the gradient of sample $z_i$. Using weights $w_i$, the gradient $G$ for the model update is given by,

$$G = \sum_{z_i \in B} w_i \cdot g(f_\theta, z_i) \quad (9)$$

### 4.1 Two-staged Reweighting for Pre-training

A common concern of pre-training is the redundancy of training data even in massive corpora mined from the web (Raffel et al., 2020). Evidently, training loss decreases exponentially early in the pre-training (Figure 3). Hence, as training proceeds, it quickly becomes the case that almost all of the computation involved in training is spent on concepts the model has already seen many times.

High SI scores indicate high gradient norms, which are critical for driving model training (McRae et al., 2022). However, encountering high SI scores from well-trained models is often a signal of noise or outlier samples (Pruthi et al., 2020). We combine the information about the property of SI scores relative to the ability of the model to calculate them *and* the nature of the pre-training data to devise a novel two-stage sample reweighting strategy. We utilize the temperature term $\tau$ when calculating the softmax weights (Equation 8) to formulate the two stages. In the first stage, which we call *'direct'* weighting, we choose $\tau > 0$, giving the data samples with higher SI scores more emphasis, driving the model learning and promoting novelty. In the second stage, or *'inverse'* weighting, where the model has matured, we use $\tau < 0$. This de-emphasizes the data samples with higher

SI scores to limit the impact of noisy and unreliable samples. Two-staged reweighting ensures that the model learns novel information early, and is able to eliminate noise at a later stage with stable learning. For temperatures $\tau_1 > 0$ and $\tau_2 < 0$, the softmax temperature at training step $i$ is given by,

$$\tau = \begin{cases} \tau_1, & i \leq I \\ \tau_2, & i > I \end{cases}, \quad (10)$$

where $I$ denotes the step where we switch stages. We refer to model-driven online sample reweighting strategy using SI scores as PRESENCE. We now adapt PRESENCE to the scale of pre-training.

## 5 Pre-training Adaptation

Pre-training requires a large batch size and is prone to instability (Krizhevsky et al., 2017). We thus adapt PRESENCE for pre-training by applying it at the microbatch level. This provides dual benefits of regularizing the pre-training while being computationally efficient.

### 5.1 Reweighting Microbatch Gradients using Self-Influence

**Microbatched Training** Microbatched training enables the use of a larger effective minibatch size. It involves dividing the full minibatch B into smaller batches, called microbatches, and individually calculating gradients for each microbatch. These gradients are then aggregated to get the minibatch gradient $G$. We present a standard microbatched training algorithm in Algorithm 1, assuming that a minibatch B is divided into $n$ microbatches, i.e. B $= \{b_i | i \in [1, n]\}$.

We first calculate the self-influence for a microbatch by replacing the individual sample $z$ with a microbatch $b$ in Equation 3 to calculate the loss. MicrobatchSI$_{\mathcal{K}}(f_\theta, \cdot)$ for microbatch $b$ is,

$$\text{MicrobatchSI}_{\mathcal{K}}(f_\theta, b) = \text{TracinSI}_{\mathcal{K}}(f_\theta, b) \quad (11)$$

| Model | Question Answering | | | Sentence Pair | Structured |
|---|---|---|---|---|---|
| | XQuAD | MLQA | TyDi QA-GoldP | XNLI | WikiAnn NER |
| Metrics | F1 | F1 | F1 | Accuracy | Span-F1 |
| *Cross-lingual zero-shot transfer (models fine-tuned on English data only)* | | | | | |
| mt5-base* | 72.92 | 64.71 | 49.23 | 74.21 | 43.28 |
| mT5-base+PRESENCE | 74.12 | 65.40 | 53.17 | 74.47 | **43.41** |
| mt5-large* | 64.15 | 51.61 | 58.75 | 67.96 | 38.66 |
| mT5-large+PRESENCE | **77.78** | **70.40** | **62.33** | **78.54** | 39.23 |
| *Translate-train (models fine-tuned on English data plus translations in target languages)* | | | | | |
| mt5-base* | 78.76 | 64.33 | 59.95 | 77.56 | 79.45 |
| mT5-base+PRESENCE | 80.44 | 65.88 | 61.75 | **80.48** | **80.50** |
| mt5-large* | 62.78 | 53.13 | 66.42 | 61.80 | 70.66 |
| mT5-large+PRESENCE | **83.15** | **70.30** | **69.04** | 79.72 | 77.26 |

Table 3: Performance comparison of mT5-base and mT5-large models pre-trained using PRESENCE with baseline pre-trained models. PRESENCE gives consistent gains over corresponding baselines. **Bold** shows the best result. * denotes our reproductions. Note that we use a batch-size of 1024 for pre-training mt5-base and a batch-size of 512 for pre-training mt5-large. Detailed discussion in Section 6.1.

| Model Variant | XQuAD | XNLI |
|---|---|---|
| mT5-large | 73.52 | 69.40 |
| mT5-large+PRESENCE | 87.44 | 88.10 |

Table 4: Results on the En-only subset for the translate-train setting. Details in Section 6.1.

To formulate microbatch level reweighting using their self-influence scores, we calculate the self-influence using Equation 11 for each microbatch to get array $\mathcal{S}$, where $|\mathcal{S}| = n$,

$$\mathcal{S} = \{s_i | s_i = \text{MicrobatchSI}_{\mathcal{K}}(f_\theta, b_i); i \in [1, n]\} \quad , \quad (12)$$

where $s_i$ denotes the SI score of $b_i$. Using the updated array $\mathcal{S}$ in Equation 6 and microbatch training strategy (Algorithm 1), we obtain the gradient for the model update $G$ using Algorithm 2 with SI based reweighting in Algorithm 2.

### 5.2 Training Setup

We use two different variants of the T5 architecture (Raffel et al., 2020), namely mT5-base and mT5-large for comparisons and pre-train on the mC4 dataset (Xue et al., 2021). We refer to our corresponding reweighted variants as mT5-base-PRESENCE and mT5-large-PRESENCE respectively. We pre-train the models with an input length 1024 and output length 229, using batch sizes of 1024 for mT5-base and 512 for mT5-large. We use loss-normalization during training with a loss normalization factor of 234496 for mT5-base and 117248 for mT5-large. For mT5-base-PRESENCE, we divide

---

**Algorithm 2** Weighted Microbatched Training

---

B ← Batch
$\mathcal{G} = \{\}$ ← Gradient array
$\mathcal{S} = \{\}$ ← Self-influence array
$G = 0$ ← Gradient initialization
$\tau$ ← weighting temperature

**for** microbatch $b_i$ **in** minibatch B **do**
    $g_i = \nabla l(f_\theta, b_i)$
    $s_i = g_i \cdot g_i$
    $\mathcal{G} \leftarrow \mathcal{G} \cup g_i$
    $\mathcal{S} \leftarrow \mathcal{S} \cup s_i$

$\mathcal{S} \leftarrow \text{normalize}(\mathcal{S})$
$\mathcal{W} = \{\text{softmax}(s_i, \tau) | s_i \in \mathcal{S}\}$
**for** $g_i, w_i$ **in** $\mathcal{G}, \mathcal{W}$ **do**
    $G \leftarrow G + g_i \cdot w_i$

---

the minibatch into $n = 8$ microbatches and for mT5-large-PRESENCE, we divide the minibatch into $n = 4$ microbatches. We select $\tau_1 = 1$, $\tau_2 = -1$, and $I = 100,000$ for the two-staged learning. We use the first layer of the encoder and first layer of the decoder as the layer set $\mathcal{K}$. We use a base learning rate of 1.0 with 10,000 warm-up steps and an inverse square root decay schedule, pre-training for 1 million steps.

## 6 Results and Analysis

### 6.1 Effectiveness of PRESENCE

We compare the performance of using PRESENCE with mT5-base (mT5-base+PRESENCE) and mT5-large (mT5-large+PRESENCE) with random pre-training in Table 3. We observe that for both variants, using PRESENCE helps improve performance on all the datasets considered. This validates the

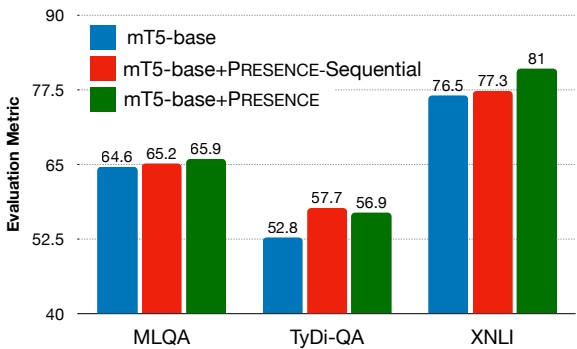

Figure 4: Performance comparison of PRESENCE-Sequential and PRESENCE with mT5-base on translate-train versions of QA and sentence pair tasks. PRESENCE is comparable or even better than PRESENCE-Sequential that uses sequential dataset filtering (Section 6.2).

| Variant | MLQA | TyDi QA | XNLI |
|---|---|---|---|
| *Cross-lingual zero-shot transfer* | | | |
| mT5-base | 61.60 | **49.09** | 69.98 |
| mT5-base+PRESENCE-D | 61.01 | 45.84 | 68.78 |
| mT5-base+PRESENCE-I | **61.88** | 42.94 | 68.37 |
| mT5-base+PRESENCE-I-D | 60.94 | 45.61 | 67.62 |
| mT5-base+PRESENCE | 61.68 | 46.47 | **70.12** |
| *Translate-train* | | | |
| mT5-base | 65.45 | 52.75 | 76.76 |
| mT5-base+PRESENCE-D | 64.98 | 55.84 | 78.80 |
| mT5-base+PRESENCE-I | 64.88 | 58.10 | 79.06 |
| mT5-base+PRESENCE-I-D | 64.65 | 55.06 | 78.96 |
| mT5-base+PRESENCE | **66.32** | **56.90** | **79.48** |

Table 5: Effect of two-staged reweighting compared to only *direct*(PRESENCE-D) or *inverse*(PRESENCE-I) weighting and inverted two-staged reweighting (PRESENCE-I-D) over mT5-base. **Bold** shows the best result. Discussions in Section 6.3.

effectiveness of PRESENCE, indicating that generating SI scores at microbatch level offers a smoother and more stable scoring approach as opposed to sample-level SI scoring. The average improvement with the PRESENCE framework is more for the large mT5 variants. A larger model (mT5-large) potentially generates more reliable SI scores when used for reweighting compared to the mT5-base model. We hypothesize these two factors as the key reasons for the significant improvements observed for the PRESENCE approach, particularly for the mT5-large model.

We also make some interesting observations for zero-shot and translate-train dataset variants. For both mT5-base and mT5-large, we observe more significant gains for PRESENCE when the training data is available in the target languages (translate-train). This indicates that reweighting might be beneficial for the model to adapt better across languages as compared to unweighted training. For instance, we consider the English subset from XQuAD and XNLI for translate-train settings. We observe that PRESENCE improves performance significantly for mt5-large experiments (Table 4).

### 6.2 Comparison with PRESENCE-Sequential

We compare PRESENCE with the multi-step PRESENCE-Sequential explained in Section 3. For a fair comparison, we pre-train mT5-base-PRESENCE for 200,000 steps on randomly sampled data and present the results in Figure 4. We observe that even though PRESENCE does not look at the complete data at once and operates in a joint online setting, it performs comparably, and in some cases, outperforms PRESENCE-Sequential. This indicates

that our online adaptation of microbatch reweighting using SI scores is competitive for model pre-training relative to sequential offline dataset filtering. One possible reason might be that the online reweighting relies upon the most recent model weights for calculating influence of training samples, providing more suitable signals for data reweighting as compared to the offline setting. Further, the joint online version forms an elegant and computationally efficient alternative to the sequential offline approach, providing an opportunity for scaling reweighting to larger models and datasets.

### 6.3 Impact of Two-staged Learning

We analyze the impact of using our two stage reweighting strategy by comparing its performance with models pre-trained purely with *direct*, i.e., $\tau = 1$ (**PRESENCE-D**) and *inverse*, i.e., $\tau = -1$ (**PRESENCE-I**) weighting. We train all the variants for 200,000 steps and compare their performance in Table 5. As shown, we observe superior performance of PRESENCE compared to the other reweighting strategies. This supports our hypothesis that pre-training probably happens in two parts: the model quickly learns new information in the first stage, after which all new information seems redundant. The second stage is important to stabilize the pre-training. To further test this, we perform reweighting in the reverse order, first performing *inverse* weighting and then *direct* weighting (**PRESENCE-I-D**). This strategy causes a degradation in the performance, as the inverse weighting initially may slow down training,

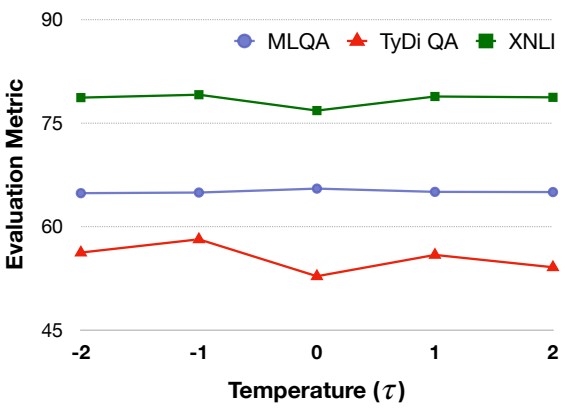

Figure 5: Effect of temperature $\tau$ during reweighting on translate-train versions of QA and sentence pair tasks. Discussions in Section 6.4.

while the direct weighting in later stages leads to increased use of noisy samples. However, there are certain datasets where either purely *direct* or *inverse* weighting perform better than PRESENCE. We believe that self-influence scores develop correlations with multilingual data based on their quantity in the corpora, which may cause varied trends in downstream tasks.

## 6.4   Scaling Microbatch Gradient Weights

Since our method currently uses two discrete values of temperature $\tau$ in Equation 8, we probe its effect on single-stage reweighting during model pre-training. These findings can be used to formulate a more granular and automated reweighting strategy using temperature scaling or continuous temperature scheduling, which we leave as future work. We pre-train models with $\tau = \{-2, -1, 1, 2\}$ for $200,000$ steps and evaluate them on the downstream tasks in Figure 5. We observe that increasing the magnitude of $\tau$ both positively and negatively affects the model performance. A possible reason might be that high positive $\tau$ leads to a large variance in the microbatch gradient weights, leading to unstable training, whereas high negative $\tau$ results in much slower convergence compared to the baselines.

## 7   Related Work

**Datasets for Pre-training LMs**   Language models are generally pre-trained on large-scale corpora scraped from the web (Devlin et al., 2019; Liu et al., 2019; Baevski et al., 2019). The most common source of obtaining large-scale data is Common

Crawl[1], a publicly-available web archive that provides "web extracted text". Raffel et al. (2020) use various heuristics such as retaining lines that end in terminal punctuations, retaining lines and pages based on a minimum number of words, etc to clean Common Crawl. Wenzek et al. (2020) use a Kneser-Ney language model (Heafield, 2011) and calculate the perplexity over training data and a high quality target domain to extract high quality documents. Multilingual pre-training has also known to depend on language and domain distributions in the corpora (Conneau et al., 2020; Du et al., 2022; Hoffmann et al., 2022). Multilingual pre-training involves an additional step of boosting low-resource language data in the corpora using temperature sampling (Conneau et al., 2020; Arivazhagan et al., 2019; Xue et al., 2021). DoReMi (Xie et al., 2023) uses a smaller proxy model to calculate domain weights of different sources comprising the mixture of the pre-training data to pre-train larger models. These works either rely on expertly crafted heuristics or require training additional models for dataset selection and filtering.

**Influence Functions and Training Data Attribution**   Influence functions help to trace a model's prediction through the learning algorithm and back to its training data, a practice commonly known as Training Data Attribution (TDA) (Guu et al., 2023). Influence functions have been extensively used in deep learning as a means of model interpretability and explainability (Linardatos et al., 2020; Arrieta et al., 2020; Guidotti et al., 2018), adversarial learning (Yuan et al., 2019; Salman et al., 2020), federated learning (Kairouz et al., 2021; Geiping et al., 2020), and identifying outliers or mislabeled samples (Koh and Liang, 2017; Yeh et al., 2018). TracIn (Pruthi et al., 2020) introduces a scalable and general first-order approximation to calculate gradient based influence, and extends it to the minibatch level. Bejan et al. (2023) formulates an automated curricular strategy using SI scores for data cleaning and filtering for NLP tasks. Influence functions have mostly been applied for supervised learning with ground truth labels for the data and have generally been explored in an offline setting.

**Data selection and online adaptation in supervised learning**   Selection functions for supervised learning often leverage training dynamics such as high loss (Jiang et al., 2019; Kawaguchi and

---

[1]https://commoncrawl.org/about/

Lu, 2020) or high prediction uncertainty (Coleman et al., 2020) to select "hard" points. Swayamdipta et al. (2020) use the change in loss over the course of training rather than each step to also eliminate noisy samples. Removing noisy training samples using offline methods is another direction for selecting training data for supervised learning (Chen et al., 2019; Pleiss et al., 2020). Paul et al. (2021) use norm of the gradient or self-influence to identify important samples early in the training to heavily prune datasets. RHO-loss (Mindermann et al., 2022) calculates a heldout loss using a proxy model and uses a combination of model loss and heldout loss to select non-noisy, non-redundant, and task-relevant samples. Ahn et al. (2023) uses per-sample gradient norm to assign importance probabilities, and trains a biased model to formulate a debiased model training strategy. Ren et al. (2018) and (Jain and Shenoy, 2022) use meta-learning for reweighting training samples within a batch for increasing robustness and selective prediction respectively. These works operate in supervised settings, requiring controlled validation sets or proxy models and adapting them to pre-training is non-trivial.

## 8    Conclusion and Future Work

We introduce PRESENCE - a method for jointly reweighting samples using self-influence (SI) scores and pre-training. We conduct an in-depth analysis of the relationship between SI scores and sample quality from a pre-training perspective and use them as a filtering objective for pre-training data selection. As sequential filtering is expensive at the scale of pre-training, we formulate PRESENCE as a joint adaptation for sample reweighting. PRESENCE outperforms baselines trained on randomly sampled and SI-filtered data on 5 datasets across 3 tasks. We believe that PRESENCE is an important first step in the research direction of data sample weighting for pre-training.

As future work, we plan to explore relationships between samples in the pre-training corpora and influence functions across languages, data sources, and domains. We also plan to formulate automated reweighting strategies using temperature scaling schedules based on the training step, training loss, and sample influence scores.

## Limitations

As a pre-training strategy, PRESENCE is computationally expensive for finding the optimal hyper-parameters, particularly for the two-staged learning. Calculation of self-influence score is only done using the first layers of the encoder and decoder for computational optimization, however, using more layers might lead to more representative weighting information. Even though we believe the training overhead of PRESENCE is significantly lesser compared to the overhead of existing methods such as sequential offline filtering, our implementation on microbatches requires a training time higher by 30% compared to training the models on randomly sampled data without any reweighting. Since the gradients across microbatches are independent, there can be ways to parallelize the computation. Our two stage training strategy currently switches at a training step which is chosen based on the total training steps, looking at the loss curve, and following how warm-up steps for learning rate schedules are decided for LM pre-training, which is ad-hoc. This can be formalized based on training loss, microbatch self-influence scores, or anchored to the dataset itself, and may lead to more suitable sample reweighting using temperature scaling.

## Acknowledgements

SC is supported by the Canada CIFAR AI Chairs program, the Canada Research Chair in Lifelong Machine Learning, and the NSERC Discovery Grant. The authors would like to thank Shubham Mittal for his assistance in compiling and analyzing the language-wise and multilingual results, and Dheeraj Rajagopal, Lucas Dixon, Pradeep Shenoy, and Kelvin Guu for insightful discussions on the work. The authors are also grateful to the reviewers and the area chairs for their helpful reviews and discussions.

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

## A Two-staged Reweighting and Learning Rate Schedulers

We create an analogy between our two stage reweighting and the transformer learning rate scheduler (Vaswani et al., 2017a). The learning rate $lr$ at step $step$ for a model with input and output dimensionality $d_{model}$ and warm-up steps $warmup$ is given by,

$$lr = d_{model}^{-0.5} \cdot min(step^{-0.5}, step.warmup^{-1.5}) \quad (13)$$

This corresponds to increasing the learning rate linearly for the first $warmup$ training steps, and decreasing it thereafter proportionally to the inverse

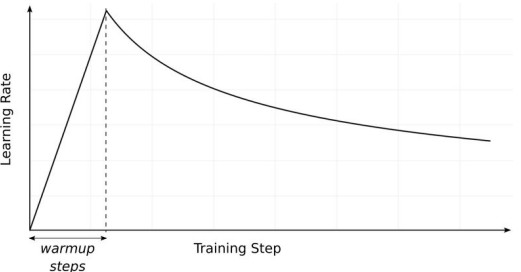

Figure 6: The learning rate scheduler as described in the Transformers (Vaswani et al., 2017b) paper. The learning rate first increases linearly for *warmup steps* and then decreases exponentially.

square root of the step number. We contrast these types of learning rate schedulers with our two stage reweighting strategy. Increasing the learning rate for a given number of steps warms up the model more by boosting the gradients, and thereafter a decay is used to enable the model to reach a minima better. Intuitively, we also aim to achieve similar learning dynamics using our two stage learning: in the first stage of learning, data samples with higher SI scores are emphasized more to drive more learning, while in the subsequent second stage, the data samples with higher SI scores are de-emphasized to limit the impact of noisy and unreliable samples while giving more weight to better quality samples and for more stable training. We believe that as future work, we can use temperature scaling schedulers inspired from learning rate schedulers to automate reweighting curricula.

## B Infrastructure

We use seqio and T5X (Roberts et al., 2022) to train our models. We use 64 TPU (Kumar et al., 2019) chips for pre-training all the models and use 8 TPU chips for fine-tuning the base variant and 16 TPU chips for fine-tuning the large variant.

## C Maturity of Models and SI Scores

Self-influence (SI) scores are calculated using the model gradients for a given objective. Their reliability, thus depends on the maturity of the model, i.e. how well the model is trained, which is being used to calculate them. Since SI scores are generally used for relative analyses, the models need not be trained till convergence. This characteristic relationship between the model's ability to predict correct labels and the reliability of SI scores becomes an important consideration when adapting SI scores for online adaptations. We have observed

that models trained for about 20% of training steps give decently reliable SI scores, however, better strategies such as choosing checkpoints where the loss decreases the most and averaging scores of multiple checkpoints have been proposed in related works (Pruthi et al., 2020). We believe that the *direct* weighting stage of PRESENCE that drives more learning also acts as an added warmup for the model's ability to predict noisy samples for the subsequent *inverse* weighting stage, enabling it to stabilize the training further. We leave the deeper analyses of SI scores on training samples early in the pre-training and in later stages as future work.