# OpenReview forum: "Self-Influence Guided Data Reweighting for Language Model Pre-training"
_EMNLP/2023/Conference — EMNLP 2023 Main_

### Official Review · Reviewer_nK5X · 2023-08-01

**Soundness:** 3

**Excitement:**

3: Ambivalent: It has merits (e.g., it reports state-of-the-art results, the idea is nice), but there are key weaknesses (e.g., it describes incremental work), and it can significantly benefit from another round of revision. However, I won't object to accepting it if my co-reviewers champion it.

**Missing References:**

Using gradient norms to sample or reweight is not a novel idea. The following paper uses the gradient norm for sampling, while the current paper uses the gradient norm for weighting the gradient. I think the two ideas are very similar, so it is worth mentioning.

@misc{ahn2023mitigating,
      title={Mitigating Dataset Bias by Using Per-sample Gradient},
      author={Sumyeong Ahn and Seongyoon Kim and Se-young Yun},
      year={2023},
      eprint={2205.15704},
      archivePrefix={arXiv},
      primaryClass={cs.LG}
}

**Paper Topic And Main Contributions:**

Paper topic
===
This paper explores the possibility of using self-influence to reweight the importance of each sample during pre-training.
They propose PRESENCE, which uses the gradient norm (equivalent to self-influence) to indicate a sample's importance and reweight the sample's gradient during backpropagation using the gradient norm. They show that the gradient norm is related to the noisiness of the training sample. By devising a two-stage reweighting schedule, they show that the mT5 pre-trained using PRESENCE is better than the model not pre-trained using PRESENCE regarding the performance on downstream cross-lingual transferring tasks.

Main Contribution
===
NLP engineering experiment

**Questions For The Authors:**

- A. Is there two-stage reweighting in Section 6.4? If no, is this experiment a reasonable ablation?
- B. Why is the performance of mT5*-large significantly worse than the performance of mT5 base for Translate-train for XQuAD, MLQA, XNLI, and WikiAnn NER in Table 3? The numbers are also significantly lower than the results from the original mT5 paper. Is there anything wrong during the pre-training in this paper's implementation?
- C. Why do we need to pre-train another model in PRESENCE-offline, instead of continually pre-training the model used for data filtering?


**Reasons To Accept:**

- Using gradient norm/self-influence for weighting the samples during pre-training is a novel idea for NLP
- The paper is generally well-written and easy to follow


**Reasons To Reject:**

- Lack of comparison to other pre-training data filtering methods.
- Significant pre-training computation overhead.
     - Detailed explanations for the above two reasons to reject: PRESENCE is essentially a data filtering method, so I expect to see the comparison with some baselines that filter the pre-training data. **The current paper only shows that PRESENCE works, but it does not show it is better than other methods**. As the authors admitted in the Limitation, PRESENCE induces 30% pre-training computation overhead. While the paper says that "`we believe the training overhead of PRESENCE is significantly lesser compared to the overhead of existing methods such as offline filtering`", this is not justified by any experiments and numbers. Moreover, offline filtering methods can be done once, and the filtered datasets can be used to train any models, while PRESENCE (the online version) needs to be applied every time when pre-training a model. This makes PRESENCE less unlikely to be applied in reality.
- Some experiment results cannot convince me that PRESENCE is good enough. The relationship between temperature $\tau$ and the downstream performance shown in Figure 5 does not seem to justify the two-stage pre-training. It seems that using a positive or negative $\tau$ can yield improvement over no reweighting. This makes me curious if it is really important to reweight using a positive or negative $\tau$. The results from Table 4 (cross-lingual zero-shot transfer) also do not show the advantage of using PRESENCE.
- It is unclear why this paper selects multilingual transferability as the downstream task for evaluation. In the mT5 paper, they also show the results on SQuAD. I wonder if PRESENCE only works for multilingual PLMs and cross-lingual transfer downstream tasks.


**Reproducibility:**

4: Could mostly reproduce the results, but there may be some variation because of sample variance or minor variations in their interpretation of the protocol or method.

**Reviewer Confidence:**

5: Positive that my evaluation is correct. I read the paper very carefully and I am very familiar with related work.

**Typos Grammar Style And Presentation Improvements:**

- I disagree with the wording in Line 283: `High SI scores result from high gradient norms`. The term "result from" is odd, since the SI used in the paper is simply gradient norm, so there is no causal relation between SI and gradient norm.
- PRESENCE-offline is a two-stage process. But this is not clearly specified before Line 242. This should be mentioned earlier in Section 3.

---

> ### Author Rebuttal · Authors · 2023-08-29
>
> We want to express our gratitude to the reviewers for their thoughtful interaction with our work and for their valuable and positive feedback. We're heartened by their recognition of the importance of the research problem (YoT7), easy to follow and understand (YoT7, nK5X), and promising results (8mAv, YoT7). We are also encouraged that our work was described as novel (nK5X), practicable (YoT7), and impactful (8mAv) with comprehensive experiments (YoT7). We've considered all the concerns that the reviewers have raised, and we've worked to address these issues as discussed below.
>
> **Reviewer's Comment: Lack of comparison to other pre-training data filtering methods**
>
> **Response:** We propose PRESENCE as an online reweighting strategy for pre-training LMs rather than a data filtration method. Please note that in the experiments in the paper, PRESENCE is applied on the corpora after the raw corpus has been filtered using numerous filtration heuristics, as described in the mT5 paper. We observe that PRESENCE is able to achieve improved performance even in this pre-filtered setting. Thus, PRESENCE is complementary to existing data filtering methods and can be used in conjunction with them.
>
> Given this difference in objectives, we feel comparison with data filtering methods is unnecessary. We shall clarify this point in the final version of the paper.
>
>
> **Reviewer's Comment: Significant pre-training computation overhead**
>
> **Response:** We point out that PRESENCE adds a 30% computation overhead (compared to the base model) compared to 100% overhead in PRESENCE-offline filtering. Moreover, in spite of this lower overhead, PRESENCE results in significant performance gains compared to other methods. In summary, PRESENCE gives more improvement while incurring lower overhead.
>
> As pointed out by the reviewer, we can have offline filtering done once for different models, but considering model-driven methods, this filtration is training dependent. Our focus in the paper is on improving performance through reweighting of a single model training, rather than amortizing the computational overhead of offline data filtering across different pretraining runs. Moreover, through our experiments, we have demonstrated the benefit of online reweighting, compared to multi-stage offline data filtering. Thank you!
>
>
> **Reviewer's Comment: Some experiment results cannot convince me that PRESENCE is good enough.**
>
> **Response:** Our goal in the paper is to present a unified framework where direct and inverse weighting can be controlled using a single parameter in the online reweighting setting. Our results do indicate that online reweighting one way or the other or a combination thereof is usually better than no weighting at all, and we establish this through extensive experiments in the paper.
>
> Through the results in Table 4, especially in the translate-train setting, we observe that PRESENCE used in combination with the two weighting phases -- Direct followed by Inverse -- results in significantly improved performance much more consistently across all datasets and tasks, as compared to just direct or inverse weighting. Since we only apply PRESENCE to pre-training, we use this to show the effectiveness of two-stage training: it is consistent as compared to just direct or inverse reweighting. Thanks!
>
> **Reviewer's Comment: It is unclear why this paper selects multilingual transferability as the downstream task for evaluation. In the mT5 paper, they also show the results on SQuAD. I wonder if PRESENCE only works for multilingual PLMs and cross-lingual transfer downstream tasks.**
>
> **Response:** PRESENCE is a general online reweighting strategy, and is not restricted to multilingual or monolingual settings. Since we used mT5 as our base model, we thought a multilingual setup would be more appropriate, as well as more representative and challenging.
>
> Having said that, please note that in our experiments we use English-only finetuning, and many of the evaluation tasks also include English as one of the target languages. We shall include this language-wise performance breakup in the final version of the paper.
>
> We hope the reviewer will recognize this multilingual evaluation as more desirable and informative rather than a limitation.
>
> **Questions**
>
> **Question: Is there two-stage reweighting in Section 6.4? If no, is this experiment a reasonable ablation?**
>
> **Response:** There is no two-stage training here as we are presenting the dependency of the temperature on the weighting. The aim of this experiment was to present a platform for future work where we discuss the possibility of automating temperature scheduling and give some initial insights into the findings. We will make this explicit.
>
>
> **Question: Why is the performance of mT5-large significantly worse than the performance of mT5 base for Translate-train for XQuAD, MLQA, XNLI, and WikiAnn NER in Table 3? The numbers are also significantly lower than the results from the original mT5 paper. Is there anything wrong during the pre-training in this paper's implementation?**
>
> **Response:** As mentioned in section 5.2, line 355, we use a batch size of 512 instead of 1024 used in the original paper due to compute reasons as pre-training is expensive, and hence the results are worse than mt5-base as well (which effectively sees double the amount of data). We will mention this in the caption of Table 3 as well to prevent any confusion for the readers. Thanks.
>
>
> **Question: Why do we need to pre-train another model in PRESENCE-offline, instead of continually pre-training the model used for data filtering?**
>
> **Response:** Thanks for the suggestion, this experiment sounds interesting. We present PRESENCE-offline not solely as a data filtering method, but rather as a step to strengthen our motivation for our online adaptation. Thus, PRESENCE-offline is not the primary method contribution and we did not perform various ablations. However, to elaborate, in PRESENCE-offline, the base model for filtering is trained on a random subset of the data. Continued pre-training of the same model on the same data will lead to weird inconsistencies as the model would have seen some data once before filtering and subsequently again after filtering. In order to avoid such inconsistencies, we train the model in the second phase from scratch. If we ensure that the dataset used to train the filtration model is completely separate from the data being filtered, we can try using it in a continued pre-training setting as well.
>
>
> **Missing References:** Thank you, this reference is relevant. We will add a discussion about this paper in the related work.
>
> **Typos Grammar Style And Presentation Improvements:** We agree with the suggestions and will incorporate the changes. Thanks!

---

### Official Review · Reviewer_YoT7 · 2023-08-12

**Typos Grammar Style And Presentation Improvements:** The article is easy to read, and the …
**Soundness:** 3

**Excitement:**

3: Ambivalent: It has merits (e.g., it reports state-of-the-art results, the idea is nice), but there are key weaknesses (e.g., it describes incremental work), and it can significantly benefit from another round of revision. However, I won't object to accepting it if my co-reviewers champion it.

**Paper Topic And Main Contributions:**

This paper introduces a novel approach to data reweighting in task-specific supervised learning and LM fine-tuning. The model-driven reweighting seems essential in the research of pre-training language models. Methods of Filtering Pre-training Data using Self-Influence and Sample Reweighting using Self-influence are relatively simple, and it appears to be effective from the experimental results.

**Reasons To Accept:**

1. Data reweighting is an important research problem for the pre-training language models, especially in today's emergence of large LM models.
2. The experimental study is comprehensive, and the results are promising.
3. The SI score is used to filter out noisy samples for pre-training dataset selection is practicable.

**Reasons To Reject:**

1. The source code has yet to be made available in the reviewed version, preventing me from verifying the effectiveness of this method through the code. I am also uncertain whether the code will be made public after accepting the paper.

2. The method's current process is undeniably intricate, entailing the optimization of multiple pipelines. These pipelines encompass sophisticated components, such as a two-phase learning strategy for online adaptation and a two-stage offline filtering procedure. By embracing this sophisticated methodology, we doubt its  potential  availability in real-world applications.

3. (minor)  More than involving three tasks and five datasets is required to verify the effectiveness of this method. I suggest expanding to more General Language Understanding Evaluation (GLUE) tasks.

**Reproducibility:**

4: Could mostly reproduce the results, but there may be some variation because of sample variance or minor variations in their interpretation of the protocol or method.

**Reviewer Confidence:**

5: Positive that my evaluation is correct. I read the paper very carefully and I am very familiar with related work.

---

> ### Author Rebuttal · Authors · 2023-08-29
>
> We want to express our gratitude to the reviewers for their thoughtful interaction with our work and for their valuable and positive feedback. We're heartened by their recognition of the importance of the research problem (YoT7), easy to follow and understand (YoT7, nK5X), and promising results (8mAv, YoT7). We are also encouraged that our work was described as novel (nK5X), practicable (YoT7), and impactful (8mAv) with comprehensive experiments (YoT7). We've considered all the concerns that the reviewers have raised, and we've worked to address these issues as discussed below.
>
> **Reviewer's Comment: The source code has yet to be made available in the reviewed version, preventing me from verifying the effectiveness of this method through the code. I am also uncertain whether the code will be made public after accepting the paper.**
>
> **Response:** Thank you for the suggestion. We would first like to highlight that implementation of PRESENCE is straightforward. The only change required in the training loop of any LM is very similar to the pseudocode following Algorithm 2 on page 6. However, we will also release the code after the acceptance. To give some background, following Algorithm 2, the core implementation involves adding about 15 lines to the trainer.py file of the t5x codebase. Thanks again for this.
>
>
> **Reviewer's Comment: The method's current process is undeniably intricate, entailing the optimization of multiple pipelines. These pipelines encompass sophisticated components, such as a two-phase learning strategy for online adaptation and a two-stage offline filtering procedure. By embracing this sophisticated methodology, we doubt its potential availability in real-world applications.**
>
> **Response:** Thank you for the comment. We would like to point out that PRESENCE (our main proposal) is a single-stage method and not a combination of offline and online stages. This only requires the optimization of the temperature hyperparameter, and we use all the other settings exactly the same as the baseline. We will clarify this more in the camera-ready version to prevent any confusion.
>
> Also, the two-phase learning strategy is inspired from the loss curve of the baseline model and we present it as an initial starting point. However, as suggested by the reviewer, we discuss this ad-hoc implementation in the limitations and have proposed using temperature schedulers as a possible future work.
>
>
> **Reviewer's Comment: (minor) More than involving three tasks and five datasets is required to verify the effectiveness of this method. I suggest expanding to more General Language Understanding Evaluation (GLUE) tasks.**
>
> **Response:** We followed this experimental setting to replicate the evaluation setup of the mt5 paper and hence chose these datasets and tasks. We believe that this evaluation is extensive considering the brevity of the paper, as it covers varied tasks such as question answering and NER along with NLI and is multilingual in nature. Thanks.

---

### Official Review · Reviewer_8mAv · 2023-08-12

**Soundness:** 4

**Excitement:**

4: Strong: This paper deepens the understanding of some phenomenon or lowers the barriers to an existing research direction.

**Paper Topic And Main Contributions:**

In this paper a framework to filter high quality data samples during pre-training process is propossed, this method uses SI score to choose the best samples in the dataset. The experiments were done on various model sizes of T5 and MC4 was used for pre-training. Results were reported for downstream tasks from XTREME benchmark.

**Reasons To Accept:**

The paper explores the area of training on high quality samples, which helps with improved performance on downstream tasks. Various sizes of the model were used for experimentation to demonstrate impact of the quality framework on the models.

**Reasons To Reject:**

The experimentation is limited to T5 based models, it will useful to carry out similar experimentation on other LMs

**Reproducibility:**

4: Could mostly reproduce the results, but there may be some variation because of sample variance or minor variations in their interpretation of the protocol or method.

**Reviewer Confidence:**

4: Quite sure. I tried to check the important points carefully. It's unlikely, though conceivable, that I missed something that should affect my ratings.

---

> ### Author Rebuttal · Authors · 2023-08-29
>
> We want to express our gratitude to the reviewers for their thoughtful interaction with our work and for their valuable and positive feedback. We're heartened by their recognition of the importance of the research problem (YoT7), easy to follow and understand (YoT7, nK5X), and promising results (8mAv, YoT7). We are also encouraged that our work was described as novel (nK5X), practicable (YoT7), and impactful (8mAv) with comprehensive experiments (YoT7). We've considered all the concerns that the reviewers have raised, and we've worked to address these issues as discussed below.
>
> **Reviewer's Comment: The experimentation is limited to T5 based models, it will useful to carry out similar experimentation on other LMs.**
>
> **Response:** Thank you for the comments. We used T5 as it is one of the most widely used LMs and it is also publicly available, making development over it easy. We thus decided to experiment with different variants of T5 rather than different LMs as they are architecturally similar (transformer-based).

---

### Meta-Review · Area_Chair_UtqM · 2023-09-20

**Recommendation:** 4

**Metareview:**

This paper investigates method for model-driven reweighting of data for LLM pre-training, and propose a two phased learning method that leverages self-influence (SI) scores as an indicator of sample importance. This paper addressed an important research problem for large language models, and proposed a novel idea with good results on using gradient norm for weighting as comments by the reviewers. On the down side, given that this paper proposed a new method for LLM pretraining with data reweighting, it can be made stronger by experimenting on other LMs in addition to the T5 based models, and running evaluations on tasks that are beyond multilingual evaluations as commented by the reviewers. The computation overhead introduced by the proposed method may limit its usage in real-world large scaling training setting.

---

### Decision · Program_Chairs · 2023-10-07

**Decision:**

Accept-Main

**Comment:**

This paper investigates method for model-driven reweighting of data for LLM pre-training, and propose a two phased learning method that leverages self-influence (SI) scores as an indicator of sample importance. This paper addressed an important research problem for large language models, and proposed a novel idea with good results on using gradient norm for weighting as comments by the reviewers. On the down side, given that this paper proposed a new method for LLM pretraining with data reweighting, it can be made stronger by experimenting on other LMs in addition to the T5 based models, and running evaluations on tasks that are beyond multilingual evaluations as commented by the reviewers. The computation overhead introduced by the proposed method may limit its usage in real-world large scaling training setting.